# Explaining the Link Between Alcohol and Homicides: Insights from the Analysis of Legal Cases in Lithuania

**DOI:** 10.3390/medicina61040657

**Published:** 2025-04-02

**Authors:** Laura Miščikienė, Justina Trišauskė, Mindaugas Štelemėkas, Kristina Astromskė

**Affiliations:** 1Health Research Institute, Faculty of Public Health, Lithuanian University of Health Sciences, Tilžės str. 18, 47181 Kaunas, Lithuania; 2Department of Health Management, Faculty of Public Health, Lithuanian University of Health Sciences, Tilžės str. 18, 47181 Kaunas, Lithuania; 3Department of Preventive Medicine, Faculty of Public Health, Lithuanian University of Health Sciences, Tilžės str. 18, 47181 Kaunas, Lithuania

**Keywords:** homicide, alcohol, victim, perpetrator, binge drinking

## Abstract

*Background and Objectives*: Alcohol consumption has been a longstanding public health concern and known link to violence. The aims of this study were to analyze alcohol-related homicide cases in Lithuania, focusing on the prevalence of binge drinking among perpetrators and victims, the situational and behavioral patterns leading to violence, and the legal outcomes of these cases. *Materials and Methods*: This study employed a retrospective analysis of court case law of criminal cases of the year 2019. The analysis was conducted by combining qualitative and quantitative analytical approaches. *Results*: The findings revealed that 84.6% of homicides occurred during binge drinking events. Alcohol intoxication was prevalent among both perpetrators (92.3%) and victims (86.5%), emphasizing the dual role of alcohol in homicide cases. Interpersonal violence was the primary pattern of homicide (78.8%), while planned homicides accounted for 21.2%. Thematic content analysis of the cases revealed that Lithuanian courts consistently regard alcohol consumption as an aggravating factor that contributes to the commission of violent crimes and influences the severity of criminal punishment. This reflects a judicial position that voluntary intoxication does not lessen legal responsibility, despite its effects on impairing judgment, heightening aggression and impulsivity, and escalating conflicts into deadly violence. *Conclusions*: Our findings revealed that the majority of alcohol-involved homicides occurred during binge drinking events, in a domestic environment, and because of unplanned acts of interpersonal violence. Targeted public health interventions should focus on strengthening alcohol control policies and enforcing stricter regulations to discourage binge drinking environments.

## 1. Introduction

In 2019, alcohol was responsible for approximately 2.6 million deaths worldwide, with 700,000 of these deaths attributed to injuries. According to the World Health Organization (WHO), alcohol consumption contributes to a substantial number of deaths globally, with a notable portion resulting from injuries, including interpersonal violence such as homicides [1]. In Lithuania, alcohol consumption has been a longstanding public health concern, with previous studies indicating alcohol prevalence in violent deaths and a link between intentional homicides and alcohol abuse [2,3].

Homicidal crimes represent one of the most severe instances of interpersonal violence, affecting individuals, families, and societies. Understanding the contributing factors to these crimes is critical for developing targeted interventions and effective prevention strategies. When considering those elements, alcohol consumption has consistently been identified as a driver of violent behavior; it can impair the judgment and escalate conflicts, increasing the likelihood of interpersonal violence [4,5,6].

Alcohol’s role in homicide extends beyond perpetration—it also influences victimization. Intoxicated individuals are more likely to place themselves in risky situations, misinterpret social cues, or be unable to defend themselves [7,8,9]. Thus, alcohol-related homicides often involve two or more intoxicated individuals, with violence escalating unpredictably in the context of impaired decision making and heightened emotional responses [10]. This dual influence of alcohol on both the perpetrator and the victim underscores the importance of analyzing alcohol-related homicide dynamics in greater detail.

Article 19 (1) of the Lithuanian Criminal Code stipulates that individuals who commit a criminal offense while under the influence of alcohol or narcotic, psychotropic, or other psychoactive substances are not exempt from criminal liability [11]. Lithuanian jurisprudence clarifies that, regardless of the degree of intoxication, voluntary substance use does not absolve an individual of criminal responsibility, as it does not eliminate culpability or legal capacity in the commission of a crime. The underlying rationale for this principle is that individuals, being aware that alcohol or other substances can significantly impair judgment, self-control, and behavioral regulation, voluntarily consume them. As a result, they enter a state in which cognitive distortions may arise, altering their perception of reality and potentially leading to a loss of behavioral control [12].

Our earlier research [3,13] revealed a high prevalence of alcohol use among both victims and perpetrators in violent crimes, particularly homicides. These findings suggest that alcohol consumption is a significant factor in fatal violent encounters. Based on this evidence, we hypothesized that most alcohol-involved homicides occur during binge drinking episodes and predominantly involve interpersonal violence rather than premeditated offenses. However, to rigorously test this hypothesis, it is essential to examine the specific circumstances surrounding these incidents—such as the context of alcohol consumption, the relationship between the individuals involved, and the situational triggers of violence.

Despite substantial research on alcohol-related violence, most existing research focuses on epidemiological data or public health policy rather than case-specific legal analysis, where circumstances surrounding homicides, and legal outcomes can be found [14,15]. In Lithuania, no comprehensive legal case analysis has been conducted to explore how alcohol contributes to homicide cases and how courts interpret these incidents. This study aims to analyze alcohol-related homicide cases in Lithuania, focusing on the prevalence of binge drinking among perpetrators and victims, the situational and behavioral patterns leading to violence, and the legal outcomes of these cases.

## 2. Materials and Methods

### 2.1. Study Design

This study employs a retrospective analysis of court case law of criminal cases from the year 2019. The data were accessed through a publicly accessible court database in Lithuania, including rulings from all Lithuanian courts [16]. District and regional courts, The Court of Appeal, and The Supreme Court of Lithuania included covering cases adjudicated between January 2019 and December 2023 to collect the latest court rulings. Official court rulings were included; these decisions contained detailed judicial reasoning, evidence assessments, and documented alcohol involvement when relevant. For cases that were appealed, only the final binding judgment was considered in the analysis. This ensured that all cases analyzed reflect the final legal outcome and judicial interpretation. The year 2019 was chosen for the analysis to reflect the situation as accurately as possible, without external circumstances, specifically the COVID-19 pandemic or any impacts of the continuous polycrisis afterwards. Also, having assessed the terms for courts decisions, possible appeal process, the case collection up to the year 2023 ensured that all rulings for homicides occurring in 2019 would be accessible.

The study was based on content analysis of the cases. The analysis was conducted by combining qualitative and quantitative analytical approaches to capture both statistical trends and nuanced judicial reasoning.

### 2.2. Data Collection

The data were collected between November 2024 and January 2025. The researchers reviewed the files and collected data using the data collection forms. Reliability was tested by having 2 researchers review the same cases. The authors entered the data into an Excel file. In the first stage of data collection, all homicide cases from Lithuanian courts (Criminal Code paragraphs 129, 130, 131) [11] that were adjudicated between January 1, 2019 and December 31, 2023 were identified (*N* = 2607). However, this total includes homicides from multiple years, while the focus of this study is exclusively on cases where the crime occurred in 2019. To ensure that only 2019 homicides were included, we applied both a time-based criterion (year of offense = 2019) and alcohol-related keywords (“alcohol”, “intoxicated”, “drinking”, “per mille”, “binge drinking”). This process resulted in a subset of 93 cases where alcohol use was explicitly mentioned in court records and the homicide occurred in 2019.

Each case was reviewed manually, and relevant information was systematically extracted into a structured database, including variables such as alcohol involvement, perpetrator and victim demographics, incident circumstances, and judicial outcomes. Cases were dismissed when alcohol consumption was possibly present but not proven. If the same case was heard in higher courts, the final effective judgements, decisions, and rulings were considered for analysis. The final count included into analysis was 52 cases, which represent homicides in 2019.

In this study, binge drinking events refer not only to instances of heavy alcohol consumption, as commonly defined in the public health literature (e.g., consuming a large quantity of alcohol within a short period) [17], but also to cases where the exact amount of alcohol consumed is unclear. In such cases, a binge drinking event is inferred based on court-established facts confirming that individuals involved in the crime had consumed alcohol. Specifically, case documentation was reviewed for explicit references to high levels of alcohol consumption, intoxication states, witness testimony regarding drinking behavior, and contextual factors (e.g., prolonged drinking sessions, multiple parties drinking together, etc.).

The study used publicly available case law, ensuring compliance with data protection and ethical research standards. No personal identifying information was collected (Table 1).

### 2.3. Statistical Analysis

Statistical analyses were performed using IBM SPSS, Version 27 for Windows. Microsoft Excel was used for data management. Statistical significance was set at <0.05. Qualitative variables are described in percentages (%) and compared using Fisher’s Exact Test. The Shapiro–Wilk Test was used to determine whether the distributions of quantitative data met the normality assumption. Approximately normally distributed quantitative variables were described by mean and standard deviation (SD), and non-normally distributed variables were described by median and 25th–75th percentiles. One-Sample Proportion Test was used to evaluate 95% CI for probability proportion.

## 3. Results

When assessing statistical data in 2019 in Lithuania, 98 homicides were registered, of which 89 were referred to the courts [18]. When assessing court decisions, we identified 52 separate cases with observed involvement of alcohol consumption.

### 3.1. Qualitative Part 

A thematic analysis of all case materials was conducted to identify recurring themes related to the discourse on alcohol consumption, particularly in the context of judicial assessments of the subjective elements of the offense and the determination of appropriate sentencing for the perpetrators.

Four main themes emerged from the content analysis of the cases (coding provided in Appendix A): (i) the influence of alcohol on the behavior and criminal act (in all cases except C14, C17, C31, C36, and C49), (ii) alcohol’s role in forming criminal intent (C1, C16, C17, C19, and C21) (iii), the impact of alcohol on the emotional state of the perpetrator (C1, C2, C6, C8, C14, C16, C18, C19, C20, C22, C27, C29, C30, C32, C35, C36, C40, C41, C42, C45, C48, and C51), and (iv) the connection between alcohol consumption and loss of control (C1, C5, C6, C7, C15, C16, C19, C20, C23, C24, C25, C27, C28, C29, C34, C35, C39, C44, C48, C50, and C51).


**Theme I: The influence of alcohol on the behavior and criminal act**


In nearly all cases, the courts established a direct link between the perpetrator’s alcohol consumption and the subsequent act of murder. The standard judicial reasoning emphasized that the offense was committed under the influence of alcohol, which significantly contributed to its occurrence:


*“[…] While under the influence of alcohol and psychotropic substances, which influenced the commission of the offense, acting as part of a group of accomplices, and without any justification, the perpetrator sought to assert themselves in front of others, displaying contempt toward them. Through insolent behaviour, they demonstrated a clear lack of respect for others. Motivated by hooliganism, they forcibly broke down the door of a dwelling and entered the apartment. […] Acting together with intent, they murdered P.V., while O.R. also threatened to kill S.J. or cause serious harm to her health.”*
(C30)

Furthermore, intoxication is considered an aggravating factor when determining the appropriate punishment, like in the case where the perpetrator inflicted fatal injuries on the victim and left without any attempt to provide aid or mitigate the harm:


*“The Court of First Instance also identified an aggravating circumstance in the case of the convicted person, P.S.: he committed the offense while under the influence of alcohol, which contributed to his criminal behaviour. […] The court also reasonably noted that P.S. struck the victim at least 12 times during a mutual conflict triggered by a trivial incident—the spilling of vodka. Despite the victim remaining conscious after sustaining injuries and attempting to clean his wounds, the convicted person made no effort to mitigate the consequences of his actions and instead left the scene.”*
(C25)

In only a few cases did the courts refrain from attributing significant influence on alcohol consumption on the perpetrator’s behavior and the criminal act. In these instances, the courts either did not consider intoxication at all or indicated that the level of intoxication was insufficient to have directly contributed to the events leading to the murder:


*“[…] It is evident from the judgment under appeal that, in establishing the aggravating circumstance, the Regional Court merely acknowledged the fact of the convicted person’s intoxication without assessing whether it had any actual influence on the commission of the offense. According to the case materials, the crime occurred on May 14, 2019, at approximately 4:45 a.m., while V.K.’s level of intoxication was tested at 5:30 a.m. on the same day. The test results indicated a low level of intoxication, measuring 0.78 g of alcohol. (Volume 1, page 15). While the fact of alcohol consumption is undisputed, the court, considering the relatively low level of intoxication and the fact that the last recorded alcohol consumption occurred five hours before the murder, concluded that the primary factor influencing the crime was O.M.’s provocative behaviour rather than V.K.’s intoxication.”*
(C3)

Also, there has been a case where a crime was committed by a person who was not intoxicated but was provoked by a heavily intoxicated victim:


*“The convicted person, while defending against a dangerous attack by F.M., who was heavily intoxicated with alcohol and drugs and had drawn a firearm during the conflict, struck the victim with a knife five times in vital areas of the body. By using a weapon and exerting intense violence, the convicted person clearly exceeded the limits of necessary self-defence.”*
(C26)

Regarding murder victims, the vast majority were also intoxicated to some degree (86.5%). Courts recognize that this factor can influence the consequences of the crime and the severity of harm suffered by the victim. However, a victim’s intoxication is not considered a mitigating factor when sentencing the offender:

“The fact that the victim was intoxicated and that this may have influenced her health or contributed to complications does not negate the causal link between the defendant’s actions and the death of Ms. A.P. The evidence in the case confirms that the violent actions of the defendant, A.M., directly led to the victim’s death. By striking the victim in the head, a vital organ, A.M. foresaw that his actions could cause serious injury, potentially leading to complications and, ultimately, death. Although he did not intend for such an outcome, he knowingly accepted the risk of it occurring.” (C24)


**Theme II: alcohol’s role in forming criminal intent**


Since intent is a fundamental criterion for establishing criminal liability, courts also consider the motives behind the intent and its nature—whether direct or indirect—when determining the most appropriate punishment for the perpetrator. In several cases, the courts have specifically addressed the moment when the intent was formed and its connection to alcohol consumption:
*“O.B. herself admitted that the idea of committing the crime arose after she had consumed alcohol”*(C1)
and
*“A.H. was under the influence of alcohol prior to the formation of the intent to commit the offense, which impaired his self-control and fundamentally distorted his assessment of the situation.”*(C16)

In most cases, the intent to commit murder was classified as indirect, largely due to the nature of the conflicts, which often arose from domestic environments, close personal relationships, and distorted perceptions of reality caused by intoxication.
*“At the same time, considering that the convicted person and D.K. were friends, that they had been drinking alcohol together, and that D.K. claimed to have injured himself upon the arrival of the ambulance, the Court of First Instance reasonably concluded that the convicted person acted with indirect intent and did not intend to kill D.K.”*(C17)
and
*“He deeply regrets what happened; he did not want to, nor did he have any intention of killing T.T. They were good friends, and everything occurred as a result of excessive alcohol consumption and a conflict.”*(C19)


**Theme III: the impact of alcohol on the emotional state of the perpetrator**


To attribute responsibility and understand the dynamics of a crime, courts assess the emotional state of perpetrators to determine whether the offense was a direct result of intoxication-related cognitive distortions or if other underlying factors—such as pre-existing animosity, provocation, or even medical conditions—played a more significant role:


*“J.J. admitted to drinking beer with his partner, L.J., prior to the incident, which led to arguments and heightened his feelings of anger and frustration. This state of emotional instability, compounded by alcohol consumption, directly contributed to his violent reaction during the conflict with L.J.”*
(C6)

Similarly:


*“The offense was motivated by anger stemming from an inappropriate relationship while intoxicated. Prior to committing the crime, the accused was habitually drunk, unemployed, and not registered with the employment service, demonstrating a disregard for social values.”*
(C20)

However, courts also consider cases where the perpetrator’s emotional state may have been influenced by alcohol consumption on the part of the victim. For instance:


*“[…] It has been indisputably established in the case that D. M. was driven to commit the crime by anger stemming from long-term psychologically traumatic relationships with his parents, his individual psychological characteristics, and the prevailing situation related to the persistent alcohol abuse of A. M. and N. M.”*
(C45)


**Theme IV: The connection between alcohol consumption and loss of control**


While perpetrators may lose control over their emotions when intoxicated, it is crucial to determine whether they were still capable of rationally understanding the consequences of their actions and exercising control over them. Courts approach this assessment with great caution, relying on forensic expert reports, as voluntary intoxication is considered an aggravating factor that does not eliminate criminal culpability:
*“The spontaneity and intoxication did not deprive the convicted person of the ability to comprehend the severity of the violent acts committed in this case. According to the forensic psychiatry and forensic psychology expert report, conducted between 29 October and 27 November 2019, A.M.’s mental and behavioural disorders resulting from alcohol consumption did not impair his capacity to understand the nature of his actions or to control them (vol. 3, pp. 2–5).”*(C23)
Similarly:
*“The forensic psychiatric and forensic psychology expert report concluded that G.M. was capable of understanding and controlling her actions at the time of the offense. Although she experienced intense anger during the commission of the crime, this emotional state did not hinder her ability to perceive and regulate her behaviour. The report further noted that the alcohol consumed, with a measured blood alcohol level of 1.87 per mille, ruled out the possibility of a physiological affective state.”*(C27)

### 3.2. Quantitative Part

The overview of various characteristics of alcohol involved homicide cases, including information on perpetrators, victims, and contextual factors, is presented in Table 2. The analysis of quantitative data of homicide cases revealed several notable aspects. The majority of incidents were predominantly committed by male perpetrators (88.5%), and most victims were male also (71.2%). During criminal incidents where alcohol consumption was involved, 86.5% of victims and 92.3% of perpetrators were intoxicated. About two third of the incidents occurred on a weekday (64.7%) and during the night (61.1%). 34.6% of criminal cases were classified as domestic, and in 30.8% of incidents, the victim was a family member. The crimes largely occurred at home (75%) and involved physical force (53.8%) or stabbing (38.5%). Interpersonal violence was the primary pattern of homicide (78.8%), while planned homicides accounted for 21.2%.

One of the most important identified results was that the majority crimes occurred during a binge drinking event—84.6%, with 95% CI for probability between 74.8% and 94.4%. Comparing the sample proportion (84.6%) of binge drinking events to the identified value of 92%, it is concluded that significantly less than 92% of cases in Lithuania end up in a binge drinking event (Z = −1.963, *p* = 0.025).

Data analysis showed considerable levels of alcohol consumption in both victims and perpetrators, with mean alcohol levels of 2.19‰ (SD 0.99) and median 1.87‰ (25th percentile—1.39 ‰, 75th percentile—2.5 ‰), respectively (Table 3). The median sentence of analyzed crimes was 10 years (Table 3).

Table 4 shows the distribution of binge drinking event by characteristics of the case. However, significant association was found only between the binge drinking event and the pattern of homicide. Binge drinking was significantly more common in cases classified as interpersonal violence (92.7%) compared to cases of premeditated homicides (54.5%).

## 4. Discussion

This study provides insights into the characteristics of alcohol-involved homicides, highlighting the significant role of binge drinking events in violent crime. Our findings revealed that 84.6% of homicides in 2019 in Lithuania occurred during binge drinking events, demonstrating the influence of alcohol consumption on lethal violence. Courts also qualify that alcohol significantly contributed to the offense occurrence. In court decisions, forensic psychiatric and psychological expert reports often highlight that perpetrators under the influence of alcohol exhibit heightened aggression and diminished impulse control, which courts identify as key factors in the escalation to violence. This underscores how alcohol is recognized not only for its physiological effects but also for its role in triggering impulsive, violent reactions in situations that might otherwise have been de-escalated. Furthermore, our analysis showed that alcohol intoxication was prevalent among both perpetrators (92.3%) and victims (86.5%), emphasizing the dual role of alcohol in homicide cases. Alcohol consumption may not only increase aggression in perpetrators but also may impair victims’ ability to de-escalate or avoid confrontation.

Notable findings included that homicides were predominantly unplanned acts of interpersonal violence (78.8%), with only 21.2% classified as premeditated crimes. This aligns with previous research indicating that alcohol-related homicides are often spontaneous, triggered by interpersonal conflicts in social or domestic settings [19,20]. And binge drinking events were more common in cases classified as interpersonal violence (92.7%) compared to premeditated homicides (54.5%). This supports the argument that alcohol-induced emotional reactivity makes individuals more likely to commit impulsive violent crimes [19,21].

Our results also indicate that most homicides occurred at home (75.0%), and the proportion of cases (34.6%) were classified as domestic homicides, which supports previous studies linking alcohol consumption to domestic violence and household conflicts [22,23]. These findings highlight the need for targeted interventions to prevent alcohol-related aggression in domestic settings.

As noted, alcohol-involved homicides are characterized as double-sided, and the relevant association identified in this study was between binge drinking events and victim intoxication. Among intoxicated victims, 88.9% were involved in a binge drinking event. Furthermore, our and other research data show that victims are often intoxicated by alcohol [3,24]. The qualitative analysis revealed that while the perpetrator’s intoxication is often regarded as an aggravating factor, the victim’s intoxication is generally not considered a mitigating circumstance when determining the offender’s punishment. Instead, courts view the victim’s alcohol consumption as part of the broader context of the crime, acknowledging its potential influence on the dynamics of the incident but without reducing the offender’s legal responsibility. This suggests that alcohol consumption is not only a risk factor for perpetrating violence but also for victimization. It aligns with previous research suggesting that alcohol as a contributing factor for homicide victimization [25,26]. This association raises important public health and policy considerations: while much attention is given to alcohol-related aggression in perpetrators, the role of alcohol in victim vulnerability also warrants further investigation. Preventive strategies should focus on reducing excessive alcohol consumption among high-risk groups, particularly in social and domestic settings.

Our study also revealed a gendered pattern in alcohol-involved homicides, with 88.5% of perpetrators and 71.2% of victims being male. This is consistent with European trends showing that men are more likely to be both perpetrators and victims of homicide ([27]). The high prevalence of male victims may suggest that many homicides arise from male-on-male confrontations, which often occur in the context of binge drinking and disputes escalated by alcohol consumption.

While this study provides valuable insights into the role of alcohol in homicide cases, it has some limitations. Firstly, the study focuses only on Lithuania, and the findings may not be directly generalizable to other countries with different drinking cultures and legal frameworks. However, our findings contribute to a broader understanding of the circumstances surrounding alcohol-involved homicides and how alcohol intoxication is treated in legal contexts, which may be useful for comparative studies. Additionally, our definition of binge drinking relied on court-reported evidence; data on alcohol intoxication levels were not always available, and thus the findings may not always accurately reflect the full extent of alcohol involvement. We analyzed homicides only for year 2019, to avoid influence of external circumstances (COVID-19), and due to the small number of cases (sample size), the possibilities for statistical analysis were limited.

This study provides insights into alcohol-involved homicides, and the combination of qualitative and quantitative methods allows for a multilayered understanding of these events. Future studies could explore comparative analyses across countries with varying alcohol control policies and taking into account the psychosocial and economic factors underlying binge drinking in homicide cases; this would provide further insights into the effectiveness of measures in reducing alcohol-related violence.

## 5. Conclusions

Our findings revealed that most alcohol-involved homicides occurred during binge drinking events and because of unintended acts of interpersonal violence. These results have several implications for alcohol control policies and homicide prevention strategies. Given the high prevalence of binge drinking in homicide cases, targeted public health interventions should focus on strengthening alcohol control policies and enforcing stricter regulations on primarily domestic binge drinking environments. Raising awareness about the risks of binge drinking and its role in violence could help reduce alcohol-related harm at the societal level.

Even though we are focusing on Lithuania as a case study, this research contributes to the more comprehensive understanding of the relationship between substance use by both victims and perpetrators and violence, homicide, and criminal liability.

## Figures and Tables

**Table 1 medicina-61-00657-t001:** Data Collection and Filtering Process.

Step	Description	Number of Cases (N)
Initial Data Collection	All homicide cases categorized under Criminal Code Paragraphs 129, 130, 131 from 1 January 2019 to 31 December 2023 were gathered.	2607
Inclusion/Exclusion Criteria Applied	Cases containing keywords: “year 2019” “alcohol”, “intoxicated”, “drinking”, “per mille”, or “binge drinking” were identified. Cases excluded if they didn’t match year of 2019, focused only on sentencing or procedural issues, or lacked documented alcohol involvement.	93
Final Case Selection	Cases were dismissed where alcohol involvement was not proven; final cases were selected after analyzing final judgments and rulings.	52

**Table 2 medicina-61-00657-t002:** Characteristics of the cases (*n* = 52).

Characteristics of the Cases		*n*	Percentages (%)
Sex of perpetrator	Male	46	88.5
	Female	6	11.5
Sex of the victim	Male	37	71.2
	Female	14	26.9
	Several victims	1	1.9
Intoxication of the victim	Yes	45	86.5
	No	7	13.5
Intoxication of the perpetrator	Yes	48	92.3
	No	4	7.7
Binge drinking event ^1^	Yes	44	84.6
	No	8	15.4
Drug use involved in the case	Yes	1	1.9
	No	51	98.1
Day of the week ^2^	Weekday	33	64.7
	Weekend	18	35.3
Time of day of the crime	Day	20	38.5
	Night	32	61.5
Victim was a family member	Yes	16	30.8
	No	36	69.2
Victim’s relationship with the perpetrator	No family connection	36	69.2
	Father	2	3.8
	Mother	1	1.9
	Partner	10	19.2
	Child	2	3.8
	Cousin	1	1.9
Domestic violence	Yes	18	34.6
	No	34	65.4
Place of the crime	Public place	12	23.1
	Home	39	75.0
	Other	1	1.9
Pattern of homicide	Interpersonal violence	41	78.8
	Premeditated	11	21.2
Lethal Force Type	Blunt Force Traum	28	53.8
	Sharp Force Trauma (stabbing, cutting)	20	38.5
	Other	4	7.7
Paragraph of criminal code	Intentional homicide	29	55.8
	Homicide with qualifying circumstances	23	44.2

^1^ Binge drinking events include not only heavy alcohol consumption occurrences, but also cases where the amount of alcohol consumed is unclear, but the circumstances established by the court confirm that all parties related to the crime consumed alcohol. ^2^ *n* = 51, in other characteristics—*n* = 52

**Table 3 medicina-61-00657-t003:** Descriptive statistics of quantitative variables.

Variable		Mean	SD
Alcohol level determined in the victim (‰) (*n* = 23)		2.19	0.99
**Variable**	**Median**		**25th–75th percentiles**
Alcohol level determined in the perpetrator (‰) (*n* = 18)	1.87		1.39–2.5
Sentence (years) (*n* = 52)	10		9–12

**Table 4 medicina-61-00657-t004:** Distribution of binge drinking events during the crime by characteristics of the case.

Characteristics of the Cases		Binge Drinking Event (“Yes”)		Binge Drinking Event (“No”)		*p* Value #	Total	
		*n*	%	*n*	%		*n*	%
Sex of Perpetrator	Male	39	84.8	7	15.2	1.0	46	100
	Female	5	83.3	1	16.7		6	
Sex of the Victim	Male	32	86.5	5	13.5	0.668	37	100
	Female	11	78.6	3	21.4		14	
Intoxication of the Victim	Yes	40	88.9	5	11.1	0.064	44	100
	No	4	57.1	3	42.9		7	
Intoxication of the Perpetrator	Yes	40	83.3	8	16.7	1.0	48	100
	No	4	100	0	0		4	
Drugs	Yes	0	0	1	100	0.154	1	100
	No	44	86.3	7	13.7		51	
Day of the week	Weekday	29	87.9	4	12.1	0.430	33	100
	Weekend	14	77.8	4	22.2		18	
Time of Day of the Crime	Day	16	80	4	20	0.695	20	100
	Night	28	87.5	4	12.5		32	
Victim was a family member	Yes	14	87.5	2	12.5	1.0	16	100
	No	30	83.3	6	16.7		36	
Domestic Violence	Yes	15	83.3	3	16.7	1.0	18	100
	No	29	85.3	5	14.7		34	
Place of the crime	Public place	10	83.3	2	16.7	1.0	12	100
	Home	33	84.6	6	15.4		39	
Pattern of homicide	Interpersonal violence	38	92.7 *	3	7.3 *	0.007 *	41	100
	Premeditated	6	54.5	5	45.5		11	
Lethal Force Type	Blunt Force Trauma	25	89.3	3	10.7	0.251	28	100
	Sharp Force Trauma (stabbing, cutting)	15	75.0	5	25.0		20	
Paragraph of Criminal Code	Intentional homicide	25	86.2	4	13.8	1.0	29	100
	Homicide with qualifying circumstances	19	82.6	4	17.4		23	

# Comparisons were performed between values of case characteristics; * statistical significance compared to “Premeditated”.

## Data Availability

Dataset available on request from the authors.

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
