# Peer review of "Explaining the Link Between Alcohol and Homicides: Insights from the Analysis of Legal Cases in Lithuania"

_medicina, 2025, doi:10.3390/medicina61040657_

Round 1
Reviewer 1 Report
Comments and Suggestions for Authors
The article in clearly written. The study combines qualitative and quantitative approaches to examine behavioral patterns, violent situations, and legal decisions. The findings emphasize that Lithuanian courts regard alcohol intoxication as an aggravating rather than a mitigating factor when determining criminal responsibility.
The study has a good interdisciplinary approach (statistical data, detailed legal analysis) with social relevance. The study presents robust methodology.
The article presents some limitations: the study is restricted to Lithuania, which reduces its applicability to countries with different legal frameworks and drinking habits; definition of binge drinking is not always clear how a binge drinking episode was determined in the analyzed court cases, the presence of alcohol is documented only in cases where it was proven in court, potentially excluding other relevant situations, 52 cases may not be sufficient to draw definitive conclusions.
Author Response
Point 1. The study is restricted to Lithuania, which reduces its applicability to countries with different legal frameworks and drinking habits.
Response 1. We acknowledge the limitation that our study is focused on Lithuania, which may affect its generalizability to countries with different legal frameworks and drinking cultures. To address this, we have added a clarification in the discussion section, p. 11 “However, our findings contribute to a broader understanding of the circumstances surrounding alcohol-involved homicides and how alcohol intoxication is treated in legal contexts, which may be useful for comparative studies.”
Point 2. The definition of binge drinking is not always clear how a binge drinking episode was determined in the analyzed court cases, the presence of alcohol is documented only in cases where it was proven in court, potentially excluding other relevant situations.
Response 2. We appreciate the reviewer’s observation and agree that the definition should be introduced. We made the changes in the Materials and Methods section p. 3-4.: “In this study, binge drinking events refer not only to instances of heavy alcohol consumption, as commonly defined in public health literature (e.g., consuming a large quantity of alcohol within a short period), but also to cases where alcohol is present, but the exact amount of alcohol consumed is unclear. In such cases, a binge drinking event is inferred based on court-established facts confirming that individuals involved in the crime had consumed alcohol. Specifically, case documentation was reviewed for explicit references to high levels of alcohol consumption, intoxication states, witness testimony regarding drinking behavior, and contextual factors (e.g., prolonged drinking sessions, multiple parties drinking together, etc.).”
Point 3. 52 cases may not be sufficient to draw definitive conclusions.
Response 3. While we agree that the number of cases may limit the ability to draw broad conclusions as in the ideal case we would need to perform a multi-year analysis of cases, but we would like to highlight that in order to select the cases that were investigating the homicides committed in 2019 we screened 2607 cases (which also included multiple proceedings of the same cases) over a period of 2019-2023. Also, each single case includes any continuation proceedings over different court instances and we still count it as the same single case. Lastly, the selected cases provides a systematic full view of alcohol related homicides committed in a one specific year in a country (even though some cases in a legal system were lasting for several years). In our view as we also emphasize in the discussion section that our study provides valuable insights into the explanation and building a bridge between the victims of homicides and perpetrators. This analysis is a piece of the puzzle of a wider analysis which analyses the autopsy data in Lithuania and perpetrators.
Reviewer 2 Report
Comments and Suggestions for Authors
I read the article with great interest. The subject, namely the correlation between alcohol consumption and murder, is well presented and the results of the study have been presented in a very clear and comprehensive manner. Overall, I think the study was well structured. The results are also very interesting, because they allow us to trace a kind of identikit of the victim and the murderer of the homicides in which alcohol consumption is involved. It would be interesting to extend the research to other countries and in the same country to repeat the study in subsequent or previous years in order to assess whether this phenomenon is affected by changes and why. For future research on the subject, I would suggest to consider among the qualitative variables also the social context of the victim and the aggressor in order to better contextualize this type of crime and evaluate the best prevention strategies.
Author Response
We sincerely appreciate the reviewer’s constructive feedback and for recognizing the significance of our research.
We acknowledge the reviewer’s suggestion to expand this research to other countries and to conduct a longitudinal analysis within Lithuania. While such extensions fall beyond the scope of the present study, we consider them important for future research.
We also appreciate the recommendation to incorporate the social context of both the victim and the perpetrator as a qualitative variable. This would offer a deeper perspective on the interplay between alcohol use and homicide, potentially leading to more targeted prevention strategies. We will consider this suggestion when designing future studies on this topic.
Reviewer 3 Report
Comments and Suggestions for Authors
The connection between alcohol and crime is an important topic, and the author's finding about the high correlation between alcohol, binge drinking by perpetrators and victims, and homicide is useful. Here are several suggestions:
- In the abstract and on p. 10 the authors state that 84.6% of 2019 homicides in Lithuania involved "binge drinking." First, the authors do not clearly define that term or explain how they determined whether drinking constituted "binge drinking." More importantly, the denominator of this statistic appears to be homicides that involved drinking, not all homicides (which according to page 3 numbered 2406, with only 52 of those cases involving alcohol use). Both of these sources of confusion must be cleared up. (The second problem also seems to afflict the data described on p. 7).
- The sample of cases is not described sufficiently. Were the authors able to get transcripts of every trial and court decision, or only appellate court decisions, or something in between?
- Cases in which both drugs and alcohol, as opposed to just drugs, should be distinguished (cf. C30, on pp. 4-5).
- Some cases are not clearly described (e.g., did C25 involve a homicide, given the victim was "cleaning" his wounds and is the description in C45 of the victim?).
- With respect to legal doctrine, the authors might want to distinguish cases in which alcohol helped create the mens rea (see C1) from those in which the mens rea was absent at the time of the offense (apparently C19). One can argue that the first defendant is much more deserving of enhanced punishment than the second. But that is not true in Lithuania?
- Information about degree of intoxication would be useful. For example, what does 1.87 mille (on p. 7, case C27) mean in terms of the metric system, the test for legal intoxication, etc.? Same type of question with respect to the information at the bottom of p. 8.
- With respect to reforms, is there any law restricting sales of alcohol in bars, etc.? (I note, however, that many of these crimes occurred in homes).
The English is pretty good. Just needs some polishing.
Author Response
Point 1. In the abstract and on p. 10 the authors state that 84.6% of 2019 homicides in Lithuania involved "binge drinking." First, the authors do not clearly define that term or explain how they determined whether drinking constituted "binge drinking." More importantly, the denominator of this statistic appears to be homicides that involved drinking, not all homicides (which according to page 3 numbered 2406, with only 52 of those cases involving alcohol use). Both of these sources of confusion must be cleared up. (The second problem also seems to afflict the data described on p. 7).
Response 1. This is a very valuable comment, and we now provided additional information on this point. We made the changes in the Materials and Methods section p. 3.: ”In this study, binge drinking events refer not only to instances of heavy alcohol consumption, as commonly defined in public health literature (e.g., consuming a large quantity of alcohol within a short period), but also to cases where the exact amount of alcohol consumed is unclear. In such cases, a binge drinking event is inferred based on court-established facts confirming that individuals involved in the crime had consumed alcohol. Specifically, case documentation was reviewed for explicit references to high levels of alcohol consumption, intoxication states, witness testimony regarding drinking behavior, and contextual factors (e.g., prolonged drinking sessions, multiple parties drinking together, etc.).” We appreciate the reviewer’s concern regarding the denominator used in our analysis and acknowledge that our initial description may have caused confusion. We have revised our manuscript to explicitly state that the denominator for alcohol-related homicide analysis is not the total number of homicides (2607) but the subset of cases where alcohol involvement was confirmed, and homicide event occurred in year 2019. We now provided additional information on this point in p. 3: “In the first stage of data collection, all homicide cases from Lithuanian courts (Criminal Code paragraphs 129, 130, 131) that were adjudicated between January 1, 2019, and December 31, 2023, were identified (N = 2,607). However, this total includes homicides from multiple years, while the focus of this study is exclusively on cases where the crime occurred in 2019. To ensure only 2019 homicides were included, we applied both a time-based criterion (year of offense = 2019) and alcohol-related keywords (“alcohol,” “intoxicated,” “drinking,” “per mille,” “binge drinking”). This process resulted in a subset of 93 cases where alcohol use was explicitly mentioned in court records and the homicide occurred in 2019.”
Point 2. The sample of cases is not described sufficiently. Were the authors able to get transcripts of every trial and court decision, or only appellate court decisions, or something in between?
Response 2. We acknowledge that our description of case selection needs to be more precise regarding the level of court decisions included. We have updated the Methods section p. 3, mentioned in Response 1 and additionally: “Official court rulings were included, these decisions contained detailed judicial reasoning, evidence assessments, and documented alcohol involvement when relevant. For cases that were appealed, only the final binding judgment was considered in the analysis. This ensured that all cases analyzed reflect the final legal outcome and judicial interpretation.”
Point 3. Cases in which both drugs and alcohol, as opposed to just drugs, should be distinguished (cf. C30, on pp. 4-5).
Response 3. In our dataset, Case C30 is the only case where both alcohol and psychotropic substances were identified. Given that this is an isolated instance, we did not introduce a separate category. However, we acknowledge that the combined influence of alcohol and drugs may have distinct effects, but that was not mentioned in court reasoning at this instance. It is also necessary to emphasize that in the case law content analysis we extracted themes depending on the argumentation of the courts. When evaluating intoxication as an aggravating circumstance in criminal cases, Lithuanian courts do not differentiate between alcohol-induced and narcotic-induced intoxication. According to the Lithuanian Criminal Code, one of the recognized aggravating factors is the commission of an offense by an individual under the influence of alcohol, narcotic, psychotropic, or other psychoactive substances, if intoxication contributed to the perpetration of the criminal act.
Point 4. Some cases are not clearly described (e.g., did C25 involve a homicide, given the victim was "cleaning" his wounds and is the description in C45 of the victim?).
Response 4. Thank you for bringing attention to the need for clearer descriptions of the case excerpts. In response, the descriptions of both cases have been enhanced. Case C25: The case does involve a homicide, despite the victim initially remaining conscious and attempting to clean his wounds. The key issue in this case was the delayed response of the perpetrator, who did not seek medical help or intervene after inflicting multiple injuries. We revised the text to clearly indicate that the victim eventually died as a result of the assault to prevent ambiguity. Case C45: In this case, D.M. was the perpetrator, not the victim. The passage refers to D.M.'s emotional distress and psychological background, which influenced their response to the situation. We adjusted, choosing another citation of the court decision to make it explicit that D.M. was the convicted person and not the victim.
Point 5. With respect to legal doctrine, the authors might want to distinguish cases in which alcohol helped create the mens rea (see C1) from those in which the mens rea was absent at the time of the offense (apparently C19). One can argue that the first defendant is much more deserving of enhanced punishment than the second. But that is not true in Lithuania?
Response 5. This is an important legal distinction, and we appreciate the reviewer’s suggestion to clarify this point. The core idea of Theme II revolves around evaluating the level of mens rea to determine the most appropriate punishment for the offender, as described in the first paragraph of Theme II description. In Lithuanian criminal case law, the Latin term mens rea is not commonly used; instead, courts assess the presence or absence of intent, distinguishing between direct and indirect intent. In all the analyzed cases, mens rea (understood as intent) is present, either in its direct or indirect form, which directly influences the severity of the punishment applied.
Point 6. Information about degree of intoxication would be useful. For example, what does 1.87 mille (on p. 7, case C27) mean in terms of the metric system, the test for legal intoxication, etc.? Same type of question with respect to the information at the bottom of p. 8.
Response 6. In the description of Table 3, we replaced the comparative term 'high levels of alcohol consumption' with 'considerable levels of alcohol consumption' to align with the Lithuanian courts' assessment of whether alcohol consumption had a relevant influence on the offense. Accordingly, in the qualitative part of the study, we present excerpts from court argumentation where the term 'low level' is used, reflecting the court’s case-specific evaluation of alcohol consumption and its significance in the outcome of the criminal case investigation. In analysis of results when referring to high levels of alcohol consumption we use a classification approach from prior studies (e.g. Afshar M, Netzer G, Murthi S, Smith GS. Alcohol exposure, injury, and death in trauma patients. J Trauma Acute Care Surg. 2015 Oct;79(4):643-8.).
Point 7. With respect to reforms, is there any law restricting sales of alcohol in bars, etc.? (I note, however, that many of these crimes occurred in homes).
Response 7. The current alcohol control policies in Lithuania do not limit sales of alcohol in bars and restaurants as the restrictions are targeting the availability of take-away alcohol only.